# Data-Driven Optimization of Polyphenol Recovery and Antioxidant Capacity from Medicinal Herbs Using Chemometrics and HPLC Profiling for Functional Food Applications

**DOI:** 10.3390/ijms27010309

**Published:** 2025-12-27

**Authors:** Vassilis Athanasiadis, Erva Avdoulach-Chatzi-Giousouf, Errika Koulouri, Dimitrios Kalompatsios, Stavros I. Lalas

**Affiliations:** Department of Food Science & Nutrition, University of Thessaly, Terma N. Temponera Str., 43100 Karditsa, Greece; vaathanasiadis@uth.gr (V.A.); eavdoulac@uth.gr (E.A.-C.-G.); ekoulouri@uth.gr (E.K.); dkalompatsios@uth.gr (D.K.)

**Keywords:** chemometric optimization, response surface methodology, multivariate analysis, HPLC-DAD, medicinal herb extracts, antioxidant activity, polyphenols, functional food ingredients, bioactive phytochemicals, green extraction processes

## Abstract

The optimization of bioactive compound extraction from medicinal herbs is critical for developing functional food ingredients with substantiated health benefits. This study employed response surface methodology (RSM) and partial least squares (PLS) regression to maximize polyphenol recovery and antioxidant capacity from five medicinal herbs (*Helichrysum stoechas*, *Chelidonium majus*, *Mentha pulegium*, *Artemisia absinthium*, and *Adiantum capillus-veneris*). A custom experimental design assessed the effects of herb identity, extraction technique, and solvent-to-solid ratio on total polyphenolic content (TPC), total flavonoid content (TFC), ferric reducing antioxidant power (FRAP), and DPPH radical scavenging activity. The PLS compromise optimum was identified for *M. pulegium* using 60% ethanol at 55 mL/g, yielding 37.54 ± 2.10 mg GAE/g dw TPC, 21.62 ± 1.15 mg RtE/g dw TFC, 334.38 ± 12.37 µmol AAE/g dw FRAP, and 262.67 ± 9.46 µmol AAE/g dw DPPH. HPLC-DAD profiling revealed 18 polyphenolic compounds (10.22 ± 0.34 mg/g dw), dominated by kaempferol-3-*O*-β-rutinoside, protocatechuic acid, and luteolin-7-*O*-glucoside. These compounds contribute complementary mechanisms: protocatechuic acid modulates oxidative and antioxidant pathways, kaempferol-3-*O*-β-rutinoside exerts cardioprotective and anti-inflammatory effects via VEGF-C binding, and luteolin-7-*O*-glucoside suppresses NF-κB-mediated inflammatory signaling. Principal component analysis (PCA) explained 84.8% of variance, clearly separating optimized from non-optimized extracts, while PLS confirmed strong correlations between specific phenolics and antioxidant indices. Overall, this integrated chemometric approach demonstrates that data-driven optimization can deliver phenolic-rich herbal extracts with robust and balanced antioxidant potential for functional food applications.

## 1. Introduction

Since prehistoric times, herbs have played an integral role in human life and society. As a primary means of preventing and treating a wide variety of diseases, its medicinal value has been documented in every ancient culture and continent. Herbal infusions, nutritional supplements, and health products commonly include herbs because of their long history of beneficial medical uses [1]. Flavonoids, phenolic acids, and tannins are only a few examples of the bioactive substances found in herbs. These compounds demonstrate potent antioxidant properties [2]. It is widely recognized that polyphenols contribute to safeguarding cells against oxidative damage and free radicals. The beneficial impact on the prevention of chronic diseases is associated with their antioxidant properties [3].

Renowned for its abundant bioactive compounds and extensive pharmacological properties, *Artemisia absinthium* is a perennial plant belonging to the Asteraceae family [4,5]. Recent clinical and experimental investigations have indicated that its extracts exhibit health-promoting activities and antioxidant properties. Sesquiterpene lactones and various polyphenolic compounds, particularly flavonoids, mediate these effects through their modulation of inflammatory pathways and potent free radical scavenging activity [6]. Recent studies have shown that *A. absinthium* is rich in bioactive molecules, rendering it an attractive candidate for the creation of anti-inflammatory and neuroprotective drugs [7].

Another Mediterranean plant from the same family, *Helichrysum stoechas*, has gained considerable knowledge for its medicinal properties. Conditions like the common cold, influenza, respiratory infections, and digestive disorders could be cured by its antibacterial, anti-inflammatory, and antidiabetic properties [8]. Flavonoids and terpenes are some of the secondary metabolites that constitute *H. stoechas*. These compounds have a significant impact on the plant’s biological activity and antioxidant properties [9].

The Papaveraceae family includes the medicinal plant *Chelidonium majus* L. among its species. It has been used as a cure for a variety of ailments, including gastrointestinal and liver issues, skin conditions, and cancer [10]. The complex profile of secondary metabolites, which includes vitamins, phenolic acids, isoquinoline alkaloids, flavonoids, and others, is responsible for its pharmacological actions [11]. These molecules have a wide range of beneficial activities for humans. Particularly important are polyphenolic metabolites, which can control inflammatory pathways and have a strong antioxidant effect. The rich bioactive profile of *C. majus*, as a whole, makes it an important medicinal plant that warrants continuous scientific study to find safe and effective therapeutic uses [12].

The Maidenhair fern, classified within the Pteridaceae family, is also known by the scientific name *Adiantum capillus-veneris*. The chemical composition of the plant varies depending on environmental factors and growing conditions. The abundance and activity of its secondary metabolites can be affected by these variables [13]. Several of the plant’s principal chemical classes—such as carotenoids, chlorophylls, phenolic acids, flavonoids, and phytosterols—have been linked to notable biological activity. The leaves of *A. capillus-veneris* have been utilized in traditional herbal medicine to treat asthma and pneumonia. They are also believed to facilitate the cleansing of the urinary system by promoting increased urine production [14]. Its treatment of scalp disorders and the stimulation of hair growth have also been documented. It can be utilized independently or in conjunction with other medicinal botanicals within both traditional and contemporary therapeutic formulations [15].

Finally, a member of the Lamiaceae family, *Mentha pulegium* L., is both aromatic and medicinal. This plant is also known as pennyroyal and is found worldwide. Its conventional applications include alleviating skin irritations and gastrointestinal disturbances. Both the fresh and dried leaves may be utilized to alleviate gastrointestinal discomfort, address ailments, and promote perspiration [16]. Essential oils and flavonoids constitute the chemical constituents of *M. pulegium*. Due to these bioactive compounds, *M. pulegium* possesses the potential for application in traditional practices as well as in future medicinal and nutraceutical products, in addition to its aromatic and bioactive properties [17]. However, it should be noted that the specific herb contains pulegone, a monoterpene, which is known to result in hepatotoxicity effects to humans, such as poor blood perfusion and hepatomegaly [18].

Response surface methodology (RSM) has been widely used to optimize bioactive compounds’ extraction conditions [19,20]. On the other hand, partial least squares (PLS) is a powerful statistical tool employed to predict a set of variables [21]. Such statistical tools could be synergistically implemented to provide optimized predictive models in food science [22]. The purpose of this research was to optimize the polyphenol recovery yield of five different medicinal plants using a combination of RSM and PLS regression. A custom design was generated for herb identity, extraction technique, and solvent-to-solid ratio. Even though several studies have highlighted the antioxidant properties of the above herbs, this study could fill the gap regarding how these statistical tools select the optimal herb/herb mixture for maximum polyphenol recovery or maximum antioxidant activity. RSM enables efficient optimization of individual responses, identifying conditions that maximize single compositional or functional attributes, whereas PLS integrates correlated responses to determine compromise optima across multiple outcomes. This combined approach is widely applied in food science because it balances compositional yield with functional performance, providing a more holistic and practically relevant optimization strategy for designing phenolic-rich extracts.

## 2. Results and Discussion

### 2.1. RSM Modeling and ANOVA

Bioactive compounds can be affected by polarity and solubility, which may render their isolation extraction a challenging task. Therefore, the extraction conditions have a substantial impact on the antioxidant capacity and extraction yield of the made extracts. Efforts to conduct experiments using fewer resources and shorter total length have lately been made by novel technology [23]. To that end, optimization of the extraction process was crucial to producing the maximum potential yield of bioactive compounds from the plants being studied. In addition, solid-to-solvent ratio has also proved to affect the extraction of selected bioactive molecules [23]. For instance, inefficient extraction can result from a low ratio, while time-consuming and decreased efficiency might be caused by a large ratio. This process was performed by using either magnetic stirring extraction or US-assisted extraction, as shown in Table 1, which grounds the reader in the raw outputs before modeling, making the variation by herb/technique obvious and justifying the need for multi-factor optimization for the five herbs.

The results of the assays by RSM modeling, in which we evaluated bioactive compounds and their antioxidant capacity, are shown in Table 1. It could be seen that the results varied for all the herbs under examination. It was observed that Chelidonium had lower recovery yields in bioactive compound extraction than the other herbs. TPC ranged from 6.6 to 53.61 mg GAE/g dw, where Chelidonium showed the lowest yield and Maidenhair fern showed the highest one. The low TPC value of Chelidonium could be a matter of maturation level of the specific herb, wherein high alkaloid levels could interfere with polyphenolic content. It was also observed that TFC ranged from 2.49 to 27.93 mg RtE/g dw, wherein Chelidonium also had the lowest yield, but Helichrysum had the highest yield. Regarding antioxidant capacity assays, Pennyroyal showed the highest values in both electron-transfer (FRAP) and radical scavenging activities (DPPH), yielding 394.76 μmol AAE/g dw and 320.22 μmol AAE/g dw, respectively. The specific bioactivity could be a matter of bioactive compounds that have low steric hindrance.

Similar results were obtained in other studies. For instance, Messaoudi et al. [24] used 80% aqueous methanol to yield 24.49 mg GAE/g extract from Pennyroyal plant, a figure close to our finding. Balciunaitiene et al. [25] yielded a lower yield than ours (10.68 mg GAE/g dw) when Artemisia plant was macerated for 2 h in 70% aqueous ethanol. In the case of *Helichrysum stoechas*, Samanidis et al. [26] conducted a comparative study of Greek Helichrysum plants, wherein the specific species yielded 76.60 mg GAE/g with ultrasound-assisted extraction and methanol as the extraction solvent. As for celandine or Chelidonium (*Chelidonium majus*), Ašimović et al. [27] employed different solvents (water, 70% aqueous methanol, and 70% aqueous ethanol) and explored the bioactive compounds of celandine leaves and stems. The results showed that leaves (13.54–57.42 mg GAE/g dw) were richer in bioactive compounds compared with stems (10.00–34.72 mg GAE/g dw). It was observed that the highest yield was obtained using 70% aqueous ethanol, and the lowest yield was reached with water as solvent, a finding that aligned with our results and highlights the impact of binary mixtures of ethanol and water. Finally, Seif et al. [28] extracted Maidenhair fern aerial parts with 50% ethanol for an hour. The authors yielded 86.16 mg GAE/g extract, a significantly higher figure than we yielded. It should be noted that such differences in yield of TPC using similar extraction conditions could be a matter of different species. The wide interval (10–100 mL/g) was selected based on pre-screening studies [24,25,28], which demonstrated saturation effects at high volumes. Curvature terms (*X*_3_^2^) reflect this non-linear behavior.

Table 2 displays the results from ANOVA for the reduced quadratic RSM models for all conducted spectrophotometric assays. The models showed high explanatory power, with R^2^ values ranging from 0.87 to 0.95 and adjusted R^2^ values from 0.77 to 0.92, indicating good fit across responses. To further assess predictive reliability, model RMSE, PRESS, and PRESS RSquare (Q^2^) values were reported. RMSE values were acceptably low (e.g., TPC RMSE = 6.42, TFC RMSE = 1.49), and Q^2^ values ranged from 0.54 to 0.93, confirming predictive validity even in cases with borderline lack of fit (e.g., FRAP *p* = 0.0953, DPPH *p* = 0.0806). These diagnostics demonstrate that the reduced quadratic RSM models are robust and suitable for subsequent optimization.

Figure 1A–E highlights the complex interplay between herb identity, extraction technique, and solvent-to-solid ratio in determining phenolic yield and antioxidant activity. The heatmaps clearly demonstrate that optimal conditions vary across species, underscoring the need for herb-specific strategies rather than a universal extraction protocol. For instance, Helichrysum achieved its maximum polyphenol yield under 60% ethanol stirring extraction at ~60 mL/g, while Chelidonium favored water extraction at 55 mL/g. Pennyroyal exhibited strong radical scavenging activity at higher solvent ratios regardless of solvent type, yet its electron-transfer capacity was enhanced by UBAE, suggesting that different extraction techniques selectively favor distinct antioxidant mechanisms. The divergence between DPPH and FRAP responses illustrates how methodological choices can shift the balance between hydrogen atom transfer and electron donation pathways. The visualization of these multivariate interactions through heatmaps facilitates rapid identification of optimal conditions and provides a natural bridge to subsequent chemometric analyses (PLS, PCA), which further confirmed the correlations between specific phenolic constituents and antioxidant indices. Overall, the results emphasize that extraction optimization must be tailored to both the botanical matrix and the desired functional property, offering valuable insights for the design of phenolic-rich herbal ingredients in functional food applications.

### 2.2. Regression Equations and Factor Effects

Regression models associated with the extraction process are shown in Equations (1)–(20). These models predict critical response variables, which include TPC, TFC, FRAP, and DPPH. The complicated interrelationships among the experimental variables are shown by the presence of interaction terms in each equation in addition to linear and quadratic ones. The presence of interactions (*X*_2_ × *X*_3_) means that the efficiency of one parameter is highly affected by the other parameter in order to maximize the yield for each assay. For instance, interrelationships were only observed in DPPH assay for all herbs, as the equations included interaction terms. Non-significant terms are not included in the models. The effects of extraction conditions and solid-to-solvent ratio were brought to light by the regression models. Quadratic terms (such as *X*_3_^2^ or *X*_2_^2^) mean that each parameter requires an optimal range, which can be either a positive or negative number. For instance, high solvent-to-solid ratio could be insufficient for the release of bioactive compounds, whereas lower values could result in reduced release due to dissolution hindrances. Extraction technique (*X*_2_) was treated as a categorical factor using contrast coding; quadratic terms are reported for model fitting but should not be interpreted as ordinal curvature.

#### 2.2.1. Total Polyphenolic Content (TPC)


(1)
$${TPC}_{Chelidonium}:2.43+21.41X_{2}+0.66X_{3}-9.38X_{2}^{2}-0.01X_{3}^{2}$$



(2)
$${TPC}_{Artemisia}:-14.66+35.99X_{2}+0.66X_{3}-9.38X_{2}^{2}-0.01X_{3}^{2}$$



(3)
$${TPC}_{Pennyroyal}:-19.95+42.48X_{2}+0.50X_{3}-9.38X_{2}^{2}-0.01X_{3}^{2}$$



(4)
$${TPC}_{Helichrysum}:-24.21+42.48X_{2}+0.81X_{3}-9.38X_{2}^{2}-0.01X_{3}^{2}$$



(5)
$${TPC}_{Maidenhair\;fern}:-24.21+42.48X_{2}+0.81X_{3}-9.38X_{2}^{2}-0.01X_{3}^{2}$$


It was found that the technique (*X*_2_) optima ranged from ~1.1 (Chelidonium) to ~2.3 (Pennyroyal, Helichrysum, Maidenhair fern). In addition, the ratio (*X*_3_) optima ranged from ~44 mL/g (Pennyroyal) to ~70 mL/g (Helichrysum/Maidenhair fern). Considering the results, different herbs match preferred extraction techniques and solvent-to-solid ratio for high TPC yield.

#### 2.2.2. Total Flavonoid Content (TFC)


(6)
$${TFC}_{Chelidonium}:-3.48+7.01X_{2}-0.01X_{3}-1.81X_{2}^{2}+0.02X_{2}X_{3}$$



(7)
$${TFC}_{Artemisia}:2.04+7.01X_{2}-0.04X_{3}-1.81X_{2}^{2}+0.02X_{2}X_{3}$$



(8)
$${TFC}_{Pennyroyal}:3.11+7.01X_{2}-0.01X_{3}-1.81X_{2}^{2}+0.02X_{2}X_{3}$$



(9)
$${TFC}_{Helichrysum}:0.81+8.94X_{2}+0.10X_{3}-1.81X_{2}^{2}+0.02X_{2}X_{3}$$



(10)
$${TFC}_{Maidenhair\;fern}:-0.25+7.01X_{2}+0.01X_{3}-1.81X_{2}^{2}+0.02X_{2}X_{3}$$


For TFC, it was observed that optima technique was 2.2–2.8 depending on herb and solvent-to-solid ratio. In addition, the interaction of *X*_2_ × *X*_3_ was positive, meaning that high technique amplifies benefits of high ratio, mainly in Helichrysum.

#### 2.2.3. Ferric Reducing Antioxidant Power (FRAP)


(11)
$${FRAP}_{Chelidonium}:39.83+1.94X_{2}+4.51X_{3}-0.03X_{3}^{2}$$



(12)
$${FRAP}_{Helichrysum}:39.83+1.94X_{2}+4.51X_{3}-0.03X_{3}^{2}$$



(13)
$${FRAP}_{Artemisia}:73.79+1.94X_{2}+4.51X_{3}-0.03X_{3}^{2}$$



(14)
$${FRAP}_{Maidenhair\;fern}:-70.38+89.92X_{2}+4.51X_{3}-0.03X_{3}^{2}$$



(15)
$${FRAP}_{Pennyroyal}:125.61+45.93X_{2}+4.51X_{3}-0.03X_{3}^{2}$$


Technique effect (*X*_2_) was of high impact for Pennyroyal/Maidenhair fern in iron-reducing assay since these equations had high multiplying factors when compared with other herbs. However, all herbs shared an optimum solvent-to-solid ratio (*X*_3_) of ~66 mL/g.

#### 2.2.4. DPPH Radical Scavenging Activity


(16)
$${DPPH}_{Chelidonium}:46.68+31.80X_{2}-0.16X_{3}-13.31X_{2}^{2}+0.21X_{2}X_{3}$$



(17)
$${DPPH}_{Artemisia}:100.74+7.37X_{2}+0.54X_{3}-13.31X_{2}^{2}+0.21X_{2}X_{3}$$



(18)
$${DPPH}_{Helichrysum}:31.91+38.80X_{2}+0.87X_{3}-13.31X_{2}^{2}+0.21X_{2}X_{3}$$



(19)
$${DPPH}_{Maidenhair\;fern}:-34.78+73.68X_{2}+0.87X_{3}-13.31X_{2}^{2}+0.21X_{2}X_{3}$$



(20)
$${DPPH}_{Pennyroyal}:234.20+31.80X_{2}+0.27X_{3}-13.31X_{2}^{2}+0.21X_{2}X_{3}$$


Regarding DPPH scavenging activity, curvature was observed in extraction technique, whereas optima varied widely among herbs. A positive interrelationship of *X*_2_ × *X*_3_ means that high technique boosts the impact of large ratios. It was observed that Helichrysum/Maidenhair fern responded strongly to both. Artemisia was found to be more ratio (*X*_3_)-driven, whereas Chelidonium and Pennyroyal peak near EtOH 60% with high ratios.

### 2.3. Multivariate Analyses (HCA, PCA, PLS)

#### 2.3.1. Hierarchical Cluster Analysis (HCA)

The integration of PCA and HCA provided a clear separation of extraction runs according to their compositional and antioxidant traits, as illustrated in Figure 2. Cluster 1, dominated by Helichrysum samples, was positioned in the upper region of the score plot and characterized by high total phenolic and flavonoid content, confirming its role as a polyphenol-rich source. Cluster 2, largely composed of Pennyroyal extracts, was separated due to the strong antioxidant capacity in FRAP and DPPH assays, highlighting that Pennyroyal, despite its moderate phenolic yield, contains compounds with disproportionately high radical scavenging efficiency. Cluster 3 represented a mixed group of herbs with consistently lower scores across all measured attributes, forming a distinct low-activity cluster. The 95% confidence ellipses further validate the robustness of these separations, emphasizing that the clustering reflects genuine biochemical differences rather than random variation. Overall, the PCA-HCA analysis demonstrates that herb identity and extraction conditions shape distinct biochemical profiles, supporting the chemometric optimization strategy and guiding the rational selection of herbs for functional food applications.

#### 2.3.2. Principal Component Analysis (PCA)

As shown in Figure 3, PCA provided deeper insights into the relationships among compositional and antioxidant variables. The model explained 84.8% of the total variance, with PC1 primarily associated with antioxidant assays (FRAP, DPPH) and PC2 aligned with phenolic and flavonoid content (TPC, TFC). The loading vectors revealed that phenolic/flavonoid content clustered along PC2, whereas antioxidant capacity clustered along PC1, reflecting distinct biochemical dimensions. The relatively low correlation between TFC and FRAP clarified the divergence observed in single-response optima among herbs, suggesting that flavonoid-rich extracts may not necessarily exhibit strong electron-transfer activity. Correlation analysis further indicated that extraction parameters *X*_2_ (technique) and *X*_3_ (solvent-to-solid ratio) were more strongly associated with PC1, particularly *X*_3_, implying that solvent ratio exerts greater influence on antioxidant outcomes than on phenolic yield. These findings highlight that extraction conditions can selectively modulate compositional versus functional attributes, reinforcing the importance of multivariate optimization strategies for designing phenolic-rich herbal extracts with targeted antioxidant properties.

#### 2.3.3. Partial Least Squares (PLS)

As shown in Figure 4, the PLS model provided a comprehensive view of how extraction parameters influence compositional and antioxidant outcomes. The 15-factor model achieved strong predictive performance (Q^2^ = 0.90 by factor 3), with variable importance in projection (VIP) scores highlighting Pennyroyal and Helichrysum (*X*_1_), solvent-to-solid ratio (*X*_3_), and its quadratic term (*X*_3_^2^) as the most influential predictors. These results confirmed that Pennyroyal and Helichrysum consistently exhibited the highest values across assays, particularly in antioxidant capacity (FRAP, DPPH). Extraction conditions were most favorable when 60% aqueous ethanol was applied under stirring, especially for maximizing total phenolic content. The solvent-to-solid ratio exerted a pronounced effect on TPC, with 55 mL/g identified as the optimal level. The correlation loading plot further demonstrated the natural grouping of extraction runs according to their chemical and antioxidant profiles, providing intuitive justification for the PLS compromise optimum. Overall, the PLS analysis validated the multivariate optimization strategy, showing that targeted tuning of herb identity and solvent ratio can deliver phenolic-rich extracts with robust antioxidant potential suitable for functional food applications.

The variance importance plot (VIP) provided quantitative evidence of the relative contribution of extraction parameters and herb identity to the PLS model. As shown in Table 3, all predictors had VIP values greater than 1, confirming their statistical significance, with scores ranging from 1.03 to 1.98. Pennyroyal (VIP = 1.98) and Helichrysum (VIP = 1.86) emerged as the most influential herb factors, reflecting their dominant roles in antioxidant and phenolic/flavonoid assays, respectively. Pennyroyal exhibited little correlation with extraction technique but consistently demonstrated superior antioxidant properties, whereas Helichrysum was strongly associated with high TPC and TFC values. The solvent-to-solid ratio (*X*_3_, VIP = 1.72) and its quadratic term (*X*_3_^2^, VIP = 1.59) were also highly impactful, particularly for optimizing TPC and FRAP responses, underscoring the importance of fine-tuning solvent levels. Interaction terms, such as Pennyroyal × ratio (VIP = 1.45), further highlighted how herb identity and extraction conditions jointly shape bioactive outcomes. Overall, the VIP analysis validated the multivariate optimization strategy by identifying the most critical predictors, thereby guiding the rational design of extraction protocols to maximize phenolic yield and antioxidant capacity. The PLS model retained three latent variables (explaining 84.8% variance). Cross-validation was performed by leave-one-out PRESS; permutation testing (n = 200) confirmed model robustness (*p* < 0.01).

### 2.4. Multi-Response Optimization

The workflow depicted in Figure 5 illustrates the progression from experimental design to multi-response optimization, contrasting the outcomes of RSM and PLS approaches. The RSM composite optimum achieved very high individual desirabilities (>0.94), but required different herbs and extraction techniques for each metric, rendering it impractical as a single extract. In contrast, the PLS compromise optimum provided a chemically feasible and balanced solution, identifying *M. pulegium* (Pennyroyal) extract with 60% aqueous ethanol at a solvent-to-solid ratio of 55 mL/g as the most effective compromise. The compromise optimum was selected because predicted and observed values aligned across all four responses (Table 4). These conditions yielded a composite desirability of 0.6474, with predicted and experimental values in close agreement.

As summarized in Table 4, RSM maximized each response independently, producing disparate optima (Helichrysum for TPC/TFC, Pennyroyal for FRAP/DPPH, and varying ratios up to 100 mL/g). The PLS compromise, however, balanced compositional and antioxidant attributes within a single extract, favoring mid-range conditions. VIP analysis confirmed that Pennyroyal, Helichrysum, solvent-to-solid ratio (*X*_3_), and its quadratic term (*X*_3_^2^) were the most influential predictors, clustering nearest to their associated responses.

Overall, the side-by-side comparison highlights a key interpretive point: the choice of optimization strategy depends on whether independence of responses (RSM) or their inter-correlation (PLS) is more relevant. For functional food applications, the PLS compromise offers a practical path toward phenolic-rich extracts with robust antioxidant potential, while RSM remains valuable for identifying theoretical maxima of individual responses.

### 2.5. HPLC Profiling of PLS Optimum

The Pennyroyal extract obtained under the PLS-optimized conditions (60% ethanol, 55 mL/g ratio) yielded a total of 18 phenolic constituents amounting to 10.22 mg/g dw, as illustrated in Figure 6 and catalogued in Table 5. HPLC-DAD profiling was performed for all five herbs; however, detailed chromatograms are shown only for the optimum extract (*M. pulegium*). It should be acknowledged as a limitation of this study that the total polyphenols quantified by HPLC-DAD (10.22 mg/g dw) were substantially lower than the spectrophotometric TPC values (37.54 mg GAE/g dw). This discrepancy is well recognized in phytochemical analysis and may arise from several factors. First, the Folin–Ciocalteu assay is non-specific and can overestimate polyphenols by detecting other reducing agents such as ascorbic acid or inorganic ions [29,30]. Second, polymerized or condensed phenolic complexes may contribute to TPC but are poorly resolved or excluded during HPLC sample preparation due to insolubility or molecular size. Third, glycosylated or bound phenolics may be underestimated by diode-array detection compared with global spectrophotometric assays. Taken together, these methodological differences explain the higher apparent values obtained by TPC. Future work employing HPLC coupled with mass spectrometry (LC-MS/MS) could provide a more comprehensive identification and quantification of complex phenolic structures, thereby reconciling spectrophotometric and chromatographic determinations. The chromatographic profile encompassed phenolic acids, flavonoid glycosides, aglycones, and diterpenes, reflecting a balanced phytochemical composition that aligns with the measured FRAP and DPPH values. Kaempferol-3-*O*-β-rutinoside (3.23 ± 0.07 mg/g dw), protocatechuic acid (1.78 ± 0.07 mg/g dw), and luteolin-7-*O*-glucoside (1.53 ± 0.04 mg/g dw) were the dominant compounds, supported by notable levels of isoharmetin-3-*O*-glucoside, rosmarinic acid, and kaempferol aglycone. Most of these compounds have also been previously documented [31,32,33] and are well known for strong radical scavenging and ferric-reducing activities, providing a mechanistic basis for the substantial antioxidant capacity observed (FRAP 334.38 ± 12.37 µmol AAE/g dw; DPPH 262.67 ± 9.46 µmol AAE/g dw). Concerning the potential toxicity of pulegone, the investigation employed a solvent of moderate-to-high polarity to preferentially extract polyphenolic compounds from Pennyroyal, thereby reducing but not entirely eliminating the possibility of co-extraction of this non-polar monoterpene. Although hydroalcoholic extracts are primarily enriched in polyphenols, trace levels of pulegone may still be present. Pulegone is hepatotoxic and subject to strict regulatory limits in foods and herbal preparations [34]. Future studies should therefore quantify any residual pulegone in hydroalcoholic extracts to ensure regulatory compliance and confirm the safety of phenolic-rich fractions intended for functional food applications.

The balanced contribution of flavonoid glycosides, phenolic acids, and flavonoid aglycones underscores the “all-rounder” antioxidant phenotype achieved by the PLS compromise, in contrast to the more extreme phenolic- or antioxidant-rich profiles derived from single-response RSM optima. This phytochemical diversity not only validates the chemometric optimization strategy but also highlights Pennyroyal as a promising candidate for functional food applications, where robust yet balanced antioxidant activity is desired.

### 2.6. Molecular Mechanisms of Selected Polyphenols

Protocatechuic acid is a representative phenolic acid with well-documented chemoprotective properties. It is known to modulate cellular signaling by up-regulating endogenous antioxidant defenses and down-regulating pro-oxidant enzymes, thereby counteracting oxidative stress [53]. For example, nicotinamide adenine dinucleotide phosphate oxidases (NOX), which are normally inactive but can generate reactive oxygen species when activated, are expressed in mammalian hearts [54]. Under treatment with the antibiotic doxorubicin, NOX enzymes donate electrons to reduce the drug to a semiquinone, producing hydrogen peroxide and superoxide anions [55,56]. Protocatechuic acid has been shown to up-regulate endogenous antioxidants, which effectively scavenge these radicals [57,58]. At the same time, pro-oxidant enzymes such as xanthine oxidase (XOD) and NOX, as well as biomarkers of lipid peroxidation like malondialdehyde, can be down-regulated in the presence of protocatechuic acid [53]. This dual regulatory activity highlights its role as a potent modulator of redox balance and supports the mechanistic basis for its contribution to the antioxidant phenotype observed in Pennyroyal extracts.

Kaempferol-3-*O*-β-rutinoside, a dominant flavonoid glycoside identified in the PLS-optimum extract, has also been extensively studied for its cardioprotective and anti-inflammatory properties. In vitro studies have demonstrated that this compound exhibits the highest protein-binding activity among kaempferol glycosides, as reported by Hu et al. [59]. Specifically, kaempferol-3-*O*-β-rutinoside interacts with vascular endothelial growth factors (VEGF), inhibiting inflammatory responses in cultured macrophages by binding to the VEGF-C receptor. This mechanism differs from conventional anti-inflammatory drugs, offering a naturally occurring alternative that may be harnessed for the prevention and treatment of inflammation-related disorders.

Taken together, the molecular mechanisms of protocatechuic acid and kaempferol-3-*O*-β-rutinoside provide a plausible biological explanation for the balanced antioxidant profile achieved by the PLS compromise. Protocatechuic acid contributes to redox homeostasis through modulation of oxidative and antioxidant pathways, while kaempferol-3-*O*-β-rutinoside enhances cardioprotective and anti-inflammatory responses. These mechanistic insights reinforce the chemometric findings and highlight that the PLS-optimized Pennyroyal extract can be translated into functional food applications, where balanced antioxidant and anti-inflammatory activity is essential for delivering substantiated health benefits.

## 3. Materials and Methods

### 3.1. Reagents and Solvents

Ethanol, methanol, hydrochloric acid, L-ascorbic acid, iron chloride hexahydrate, sodium hydroxide, DPPH (1,1-diphenyl-2-picrylhydrazyl), 2,4,6-tris(2-pyridyl)-s-triazine (TPTZ), and all HPLC chemical standards were from Sigma-Aldrich (Steinheim, Germany). The Folin–Ciocalteu reagent, gallic acid, and anhydrous sodium carbonate were purchased from Penta (Prague, Czech Republic). A deionizing column provided the deionized water used for the conducted experiments.

### 3.2. Plant Materials

Dried herbs were used for all experiments. Artemisia (*Artemisia absinthium*), Chelidonium (*Chelidonium majus*), Helichrysum (*Helichrysum stoechas*), and Maidenhair fern (*Adiantum capillus-veneris*) were bought from a local shop in Xanthi, Greece. Pennyroyal (*Mentha pulegium*) was purchased from a local shop in Makrinitsa, Greece. The dried herbs were ground using a laboratory blender to obtain a fine powder. The powders were then sifted through a 40-mesh sieve (<400 μm) to remove coarse or unwanted particles. All powdered samples were stored in a freezer (−40 °C) until further analysis.

### 3.3. Instruments and Statistical Software

The lyophilization process was conducted through a BioBase BK-FD10P freeze-dryer (BioBase, Jinan, China). An electric mill (SilverCrest Coffee Grinder SKME150; Kompernass Handels GmbH, Bochum, Germany) was employed to reduce the particle size of herbs and increase the surface area of the powders, which were separated by means of a sieving process using an Analysette 3 PRO (Fritsch GmbH, Oberstein, Germany) apparatus. The stirred-tank extraction process was feasible using stirring hotplates from Heidolph Instruments GmbH & Co. KG (Schwabach, Germany). The ultrasonication (US) process was performed using an Elmasonic P70H US sonication bath from Elma Schmidbauer GmbH (Singen, Germany). The supernatant liquid from the resulting extract was separated via a centrifugation process and a NEYA 16R centrifuge from Remi Elektrotechnik Ltd. (Palghar, India). The extracts were finally stored in a freezer at −40 °C; the model was Platinum 500 from Angelantoni Life Sciences (Massa Martana, Italy).

Spectrophotometric determinations were conducted using a Shimadzu UV-1900i UV/VIS spectrophotometer (Kyoto, Japan). Polyphenolic compound separation, identification, and quantification were feasible through a Shimadzu CBM-20A high-performance liquid chromatography system equipped with an SPD-M20A diode array detector (Shimadzu Europa GmbH, Duisburg, Germany). The detector was loaded with a Phenomenex Luna C18(2) column (100 Å, 5 μm, 4.6 mm × 250 mm) (Torrance, CA, USA). Detection wavelengths were set at 280, 320, and 360 nm to cover phenolic acids and flavonoids.

Statistical analysis was performed through JMP^®^ Pro 16 software (SAS, Cary, NC, USA). The analysis also included response surface methodology (RSM) and distribution analysis.

### 3.4. Experimental Design

A custom D-optimal design was employed to optimize three process factors for their influence on four responses: total polyphenolic content (TPC), total flavonoid content (TFC), ferric reducing antioxidant power (FRAP), and DPPH radical scavenging activity. The factors under investigation were the following: *X*_1_ is the herb identity, a five-level categorical factor comprising Helichrysum, Pennyroyal, Maidenhair fern, Chelidonium, and Artemisia; *X*_2_ is the extraction technique, a three-level ordinal factor coded numerically as 1 for water extraction (maceration at 80 °C for 5 min), 2 for ethanol 60% extraction (60 °C for 90 min under stirring process), and 3 for ultrasound-bath-assisted extraction (UBAE, ultrasonication at 37 kHz, 80% amplitude, for 20 min), with quadratic term included, acknowledging its limitations; and *X*_3_ is the solvent-to-solid ratio, treated as a continuous variable spanning from 10 to 100 mL/g. For statistical purposes, *X*_3_ was coded to a dimensionless variable *z*_3_ in the range −1 to +1 using the transformation *z*_3_ = (*X*_3_ − 55)/45, with 55 mL/g as the design center (0).

The custom design was generated using a coordinate-exchange algorithm to efficiently estimate the main effects of all three factors, the quadratic terms for *X*_2_ (as an ordinal factor) and *X*_3_, and the two-factor interactions *X*_1_ × *X*_2_, *X*_1_ × *X*_3_, and *X*_2_ × *X*_3_, while preserving model hierarchy. The final design comprised 23 unique treatment combinations, with five design points replicated to provide pure error estimates for lack-of-fit testing. Experimental runs were executed in a fully randomized order to minimize systematic bias, and no blocking was applied. All factor settings were confined within the specified ranges and levels, and subsequent optimization did not extrapolate beyond this experimental region.

Separate reduced second-order models were fitted for each of the four responses. For modeling, the five-level categorical herb factor (*X*_1_) was coded via orthogonal contrasts, *X*_2_ was treated as an ordinal numeric variable to permit estimation of curvature (*X*_2_^2^), and *X*_3_ was used in its coded form (*z*_3_). Stepwise regression with hierarchy preservation was used to eliminate non-significant terms, producing parsimonious models for each response. Model adequacy was assessed using ANOVA, lack-of-fit testing against pure error, residual diagnostics, and fit statistics (R^2^, adjusted R^2^).

The general form of the fitted second-order model for a given response *Y* (TPC, TFC, FRAP, or DPPH) was(21)$$Y=\beta_{0}+\sum_{l=1}^{L\text{-}1}\beta_{1l}C_{1l}+\beta_{2}X_{2}+\beta_{22}X_{2}^{2}+\beta_{3}z_{3}+\beta_{33}z_{3\;}^{2}+\sum_{l=1}^{L\text{-}1}\gamma_{l2}C_{1l}X_{2}+\sum_{l=1}^{L\text{-}1}\gamma_{l3}C_{1l}z_{3}+\beta_{23}X_{2}z_{3}$$
where *Y* was the response variable (TPC, TFC, FRAP, DPPH); *β*_0_ and *β* were the model coefficients (intercept, linear, quadratic, interaction effects); *C*_1*l*_ was the coded contrasts for herb factor *X*_1_ (Helichrysum, Pennyroyal, etc.); *X*_2_ was the extraction technique code (1 = water, 2 = EtOH 60%, 3 = UBAE); and *z*_3_ was the coded solvent-to-solid ratio: (*X*_3_ − 55)/45.

### 3.5. Bioactive Compounds’ Determinations

#### 3.5.1. Total Polyphenols Analysis

Total polyphenol content (TPC) was evaluated using a photometric methodology involving the widely known Folin–Ciocalteu procedure [60]. Accurate volumes of a properly diluted sample extract were mixed with the same volume of the yellowish Folin–Ciocalteu reagent (500 μL) and were left for 2 min to equilibrate. Right after, 5% *w*/*v* aqueous sodium carbonate was inserted to fill the 5 mL volumetric flask, and the mixture was incubated for 20 min at 40 °C. The absorbance at 740 nm was recorded, and the TPC was calculated using calibration curve (shown in Table A1). The results were expressed as mg gallic acid equivalents (GAE) per g of dry weight (dw).

#### 3.5.2. Total Flavonoids Analysis

An established methodology [61] was employed to measure total flavonoid content (TFC). An accurate volume (500 μL) of a properly diluted sample was inserted into 200 μL of a mixture consisting of aluminum chloride (5% *w*/*v*) and sodium acetate (0.5 M). A 5 mL volumetric flask was filled with aqueous ethanol (35% *v*/*v*), and the final mixture was kept devoid of light for 30 min. The absorbance at 415 nm was recorded immediately; therefore, TFC was calculated using a calibration curve (Table A1). The results were expressed as mg of rutin equivalents (RtE)/g dw.

#### 3.5.3. Individual Polyphenols Quantification

Further evaluation of TPC and TFC was conducted by quantifying individual polyphenols through chromatographic determination, as previously documented [61]. The separation of compounds was performed in a column that had the temperature set at 40 °C. The mobile phase had a constant flow rate of 1 mL/min and consisted of solutions A (0.5% *w*/*v* formic acid in water) and B (0.5% formic acid *w*/*v* in acetonitrile). The gradient program initiated from 0% to 40% B in a 10 min span, increased to 50% B in 10 min, then 70% B in 10 min, and finally remained constant at 70% B for 10 min. The identification of individual polyphenols was feasible by comparing the retention times of chemical standards with separated compounds. The quantification of these compounds was performed using calibration curves (Table A2). The results were initially calculated in mg/L of extract and subsequently expressed in mg/g dw.

### 3.6. Antioxidant Capacity Assays

#### 3.6.1. Electron-Transfer Antioxidant Activity

The electron-transfer antioxidant activity of extracted compounds was determined using ferric-reducing antioxidant power (FRAP) assay, as previously established [62]. This assay involves the reduction in iron ions and the formation of a stable blueish complex (Fe^+2^–TPTZ), the absorbance of which peaked at 620 nm. Briefly, a volume of 500 μL of properly diluted extract was inserted into a 5 mL volumetric flask along with the same volume of iron (III) chloride solution (in 0.05 M HCl). After incubation at 37 °C for 30 min, the volume was filled with the solution consisting of TPTZ ligand (1 mM dissolved in 0.05 M HCl) and was left to equilibrate for 5 min. A calibration curve (shown in Table A1) was used to calculate the iron-reducing capacity of extracts, whereas the results were expressed as μmol of ascorbic acid equivalents (AAE)/g dw.

#### 3.6.2. Radical Scavenging Activity

The radical scavenging activity of extracted compounds was evaluated using a well-known stable compound (DPPH^•^) [62]. The scavenging capacity of extracted antioxidant compounds that led to the decolorization of this radical was the rationale behind this assay. An accurate volume of 125 μL of herb extracts was mixed with the appropriate volume of the purple DPPH^•^ solution (100 μM in methanol) to fill a 5 mL volumetric flask. The delocalization of *π* electrons resulted in the decolorization of the stable DPPH^•^, wherein the initial and final absorbance (after 30 min storage in the darkness) was spectrophotometrically evaluated at 515 nm. This procedure also involved a blank solution where 125 μL of methanol was used instead of the sample. Antiradical activity (*A*_AR_) assessment was performed using an ascorbic acid calibration curve, as shown in Table A1. The results were expressed as μmol AAE/g dw.

### 3.7. Statistical Analysis

An independent and reduced second-order polynomial model was fitted in the response surface methodology (RSM) framework for each response variable (i.e., TPC, TFC, FRAP, and DPPH assays). Based on the fitted RSM models, heatmaps were created to illustrate the factor-response connections. We used effect summaries and Pareto plots to examine the impact size and ranking. Principal component analysis (PCA) was used to evaluate variance structure and trait correlations, while hierarchical cluster analysis (HCA) with Ward’s method and Euclidean distances was used to explore cluster patterns among the 23 experimental runs. Each of the four responses was modeled concurrently in latent variable space using partial least squares (PLS) regression, with predictor relevance evaluated using variable importance in projection (VIP) scores.

A composite desirability function was constructed using the independent RSM models (equal weights, larger is better), and optimization in the PLS profiler was used to determine a latent space compromise setting. Two approaches were used in the multi-response optimization process. The extract was chemically profiled using HPLC-DAD, and experimental validation was carried out for the PLS identified optimal. All statistical tests were all two-sided and used a *p*-value threshold of <0.05 as the significance level.

## 4. Conclusions and Practical Implications

In conclusion, this study delineated two distinct high-performance phenotypes for enhanced botanical extraction. The first phenotype was a phenolic/flavonoid-rich profile obtained using Helichrysum as plant material, 60% ethanol as extraction solvent, and a high solvent-to-material ratio (70–100 mL/g). The second phenotype was an antioxidant-rich profile produced by *M. pulegium* (Pennyroyal) under ultrasound-bath-assisted extraction (UBAE, 37 kHz, 80% amplitude, 20 min) at a comparable ratio (~55 mL/g). While these single-response optima highlight the potential of specific herbs under tailored conditions, curvature effects revealed that extreme settings are sub-optimal for at least one important metric, underscoring the need for a balanced approach. Future studies should also address the potential co-extraction of the hepatotoxic monoterpene pulegone, ensuring quantification and compliance with EFSA regulatory limits.

The PLS compromise optimum emerged as a practical “all-around” solution, selected because predicted and observed values aligned across all four responses. This approach offered balanced bioactivity across compositional and antioxidant indices. For practical applications, extraction conditions can be tuned to target specific chemical classes: phenolic/flavonoid recovery is maximized with Helichrysum under 60% ethanol and high ratios (70–100 mL/g), whereas antioxidant capacity is optimized with Pennyroyal under UBAE at ratios above 74 mL/g. A balanced profile, suitable for functional food applications, was achieved with Pennyroyal extract in 60% ethanol at a mid-range ratio (55 mL/g).

Overall, the methodological framework presented here—integrating RSM, PCA/HCA, PLS, and HPLC profiling—provides a robust foundation for the optimization of botanical extractions. Beyond the specific herbs studied, this workflow demonstrates originality by combining univariate and multivariate chemometrics with phytochemical profiling, and can be adapted to other plant matrices or scaled for industrial applications. This strategy guides the rational design of phenolic-rich extracts with tailored bioactivity profiles, while ensuring safety and regulatory compliance.

## Figures and Tables

**Figure 1 ijms-27-00309-f001:**
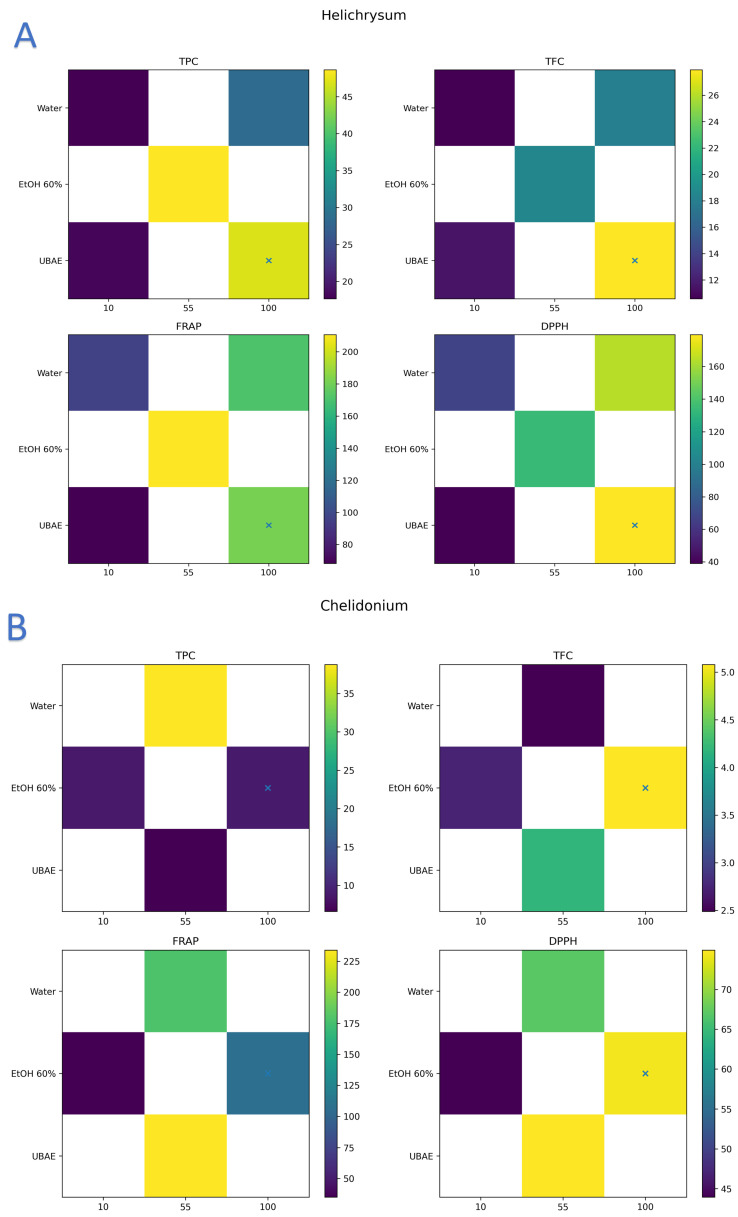
Heatmap visualization of single-response optima for five medicinal herbs: (**A**) Helichrysum, (**B**) Chelidonium, (**C**) *M. pulegium* (Pennyroyal), (**D**) Artemisia, and (**E**) Maidenhair fern, across three extraction techniques and solvent-to-solid ratios. Each subfigure (**A**–**E**) presents four biochemical response variables: total polyphenolic content (TPC), total flavonoid content (TFC), ferric reducing antioxidant power (FRAP), and DPPH radical scavenging activity. Color gradients represent response intensity, with warmer tones indicating higher values. The “×” marker within each heatmap denotes the PLS-derived optimum, corresponding to the maximum composite desirability value identified by the multivariate model. *X*_1_ corresponds to herb identity, *X*_2_ to extraction technique (1 = water, 2 = 60% ethanol, 3 = UBAE), and *X*_3_ to solvent-to-solid ratio (10, 55, 100 mL/g). This herb-specific layout highlights distinct bioactivity profiles and facilitates the identification of optimal extraction conditions.

**Figure 2 ijms-27-00309-f002:**
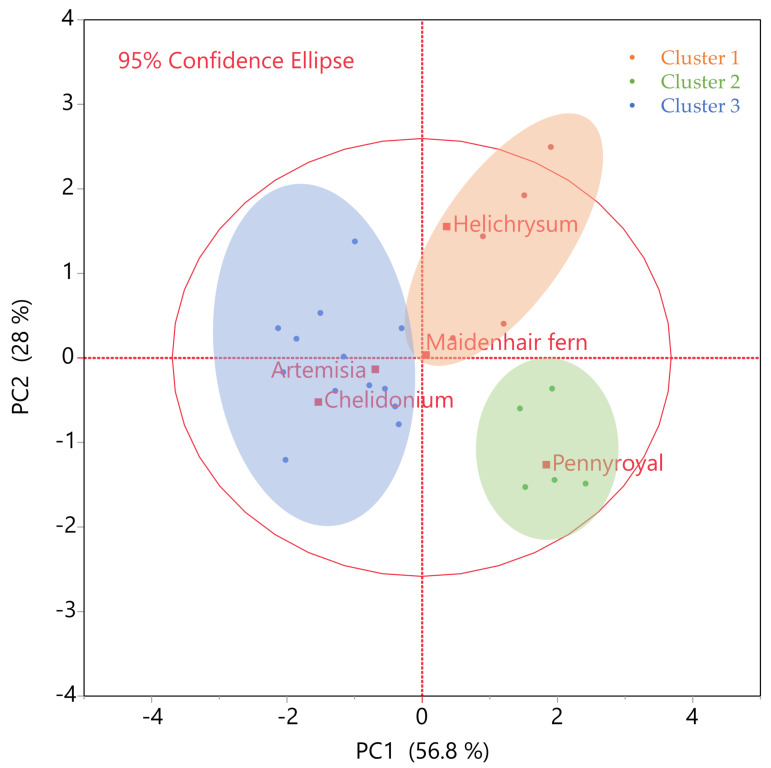
Principal component score plot (PC1 vs. PC2) with hierarchical cluster analysis (HCA) overlay. Data points are color-coded according to HCA clusters: Cluster 1 (orange, phenolic/flavonoid-rich: Helichrysum, Maidenhair fern), Cluster 2 (green, antioxidant-rich: Pennyroyal), and Cluster 3 (blue, low-activity mixed group: Chelidonium, Artemisia). Axes represent the first two principal components (PC1: 56.8%, PC2: 28%), capturing 84.8% of total variance. Ellipses indicate the 95% confidence regions for each cluster, highlighting compositional and functional separation among extracts.

**Figure 3 ijms-27-00309-f003:**
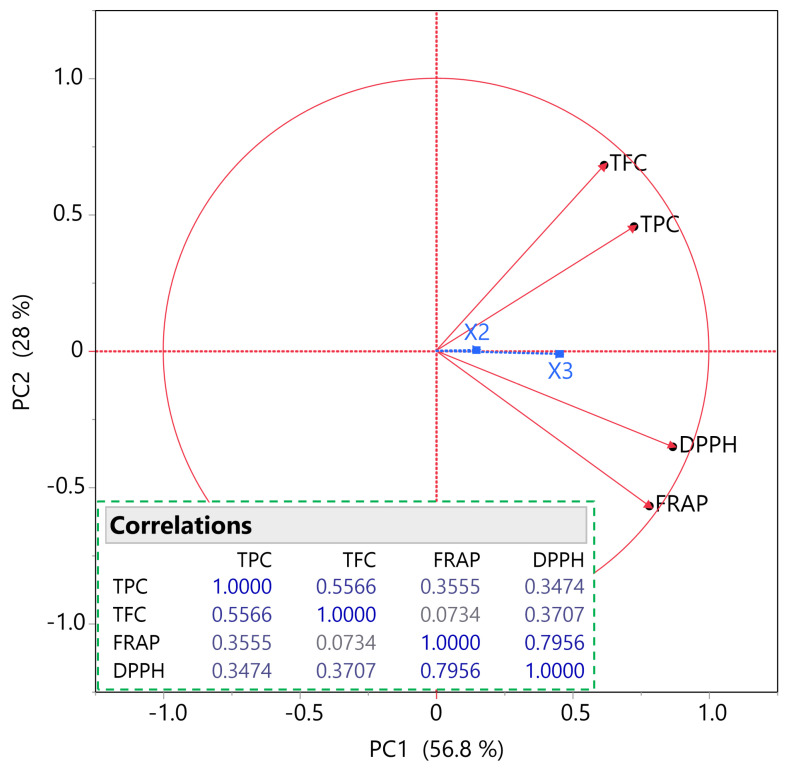
PCA loading plot and Pearson correlation matrix of the response variables (TPC, TFC, FRAP, DPPH). Red vectors indicate the contribution and orientation of each variable in the PCA space, with PC1 and PC2 explaining 56.8% and 28.0% of the total variance, respectively. The correlation circle illustrates variable interrelationships, while the accompanying matrix summarizes pairwise Pearson correlation coefficients among compositional and antioxidant indices.

**Figure 4 ijms-27-00309-f004:**
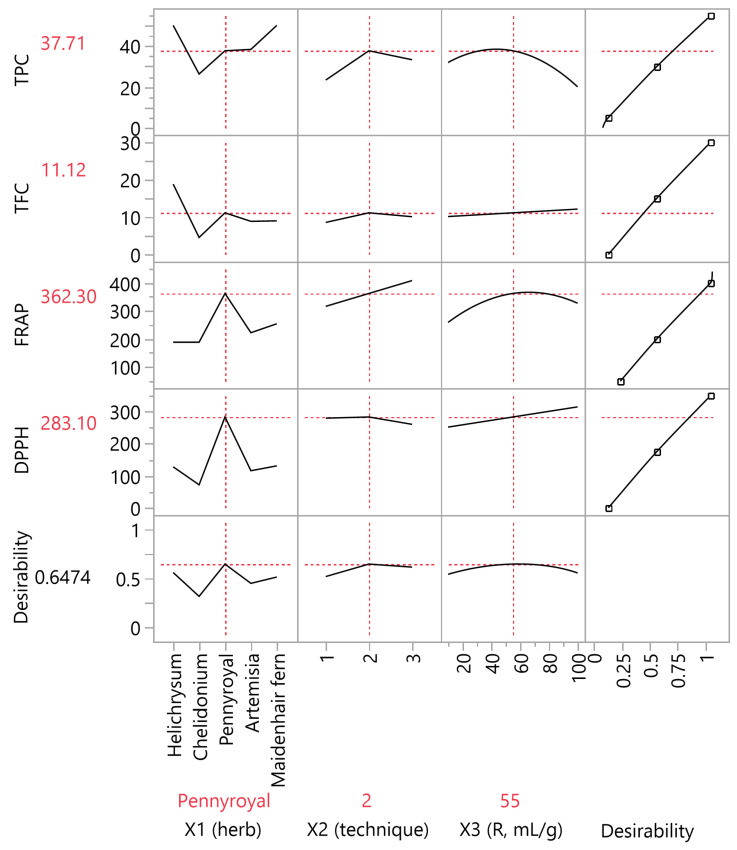
Partial least squares (PLS) correlation loading plot from the 15-factor model, illustrating relationships between predictors (*X*_1_: herb identity, *X*_2_: extraction technique, *X*_3_: solvent-to-solid ratio including polynomial and interaction terms) and responses (TPC, TFC, FRAP, DPPH) in latent variable space. Red dashed lines indicate the optimal predictor levels associated with a composite desirability score above the significance threshold of 0.80. The plot highlights variable contributions and clustering patterns that support the selection of compromise extraction conditions.

**Figure 5 ijms-27-00309-f005:**
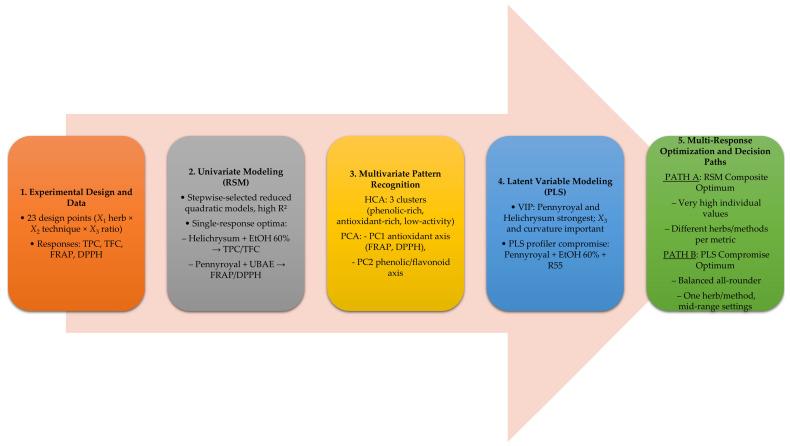
Workflow from experimental design to RSM/PLS optima and decision paths. The diagram summarizes the sequential steps of experimental design, univariate modeling (RSM), multivariate pattern recognition (PCA/HCA), latent variable modeling (PLS), and multi-response optimization, highlighting the contrast between RSM composite optima and PLS compromise solutions.

**Figure 6 ijms-27-00309-f006:**
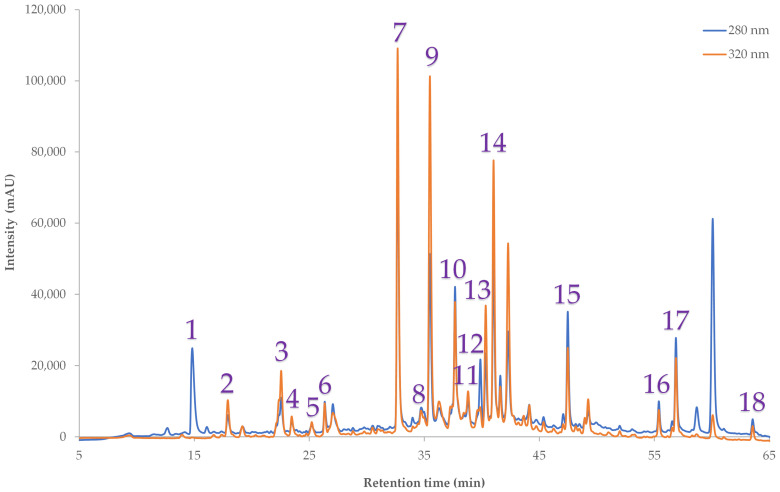
Representative HPLC chromatograms at 280 nm and 320 nm of the PLS-optimum extract (Pennyroyal, EtOH 60%, solvent-to-solid ratio 55), showing the identified phenolic compounds. Peak numbers correspond to those listed in Table 5.

**Table 1 ijms-27-00309-t001:** Experimental results showing the effects of the three independent variables—herb identity (*X*_1_), extraction technique (*X*_2_), and solvent-to-solid ratio (*X*_3_)—on the dependent variables: total phenolic content (TPC), total flavonoid content (TFC), ferric-reducing antioxidant power (FRAP), and DPPH radical scavenging activity.

Design Point	*X*_1_—Herb	*X*_2_—Technique	*X*_3_—Ratio (mL/g)	TPC (mg GAE/g dw)	TFC (mg RtE/g dw)	FRAP (μmol AAE/g dw)	DPPH (μmol AAE/g dw)
1	Maidenhair fern	EtOH 60%	55	53.61	9.29	285.09	137.18
2	Pennyroyal	Water	55	22.25	8.42	277.63	274.50
3	Chelidonium	Water	55	38.78	2.49	178.65	67.04
4	Chelidonium	UBAE	55	6.60	4.19	233.97	74.93
5	Helichrysum	Water	100	28.47	18.09	170.64	164.64
6	Chelidonium	EtOH 60%	100	8.89	5.08	107.52	74.39
7	Artemisia	UBAE	10	17.76	7.02	151.64	21.01
8	Pennyroyal	EtOH 60%	10	35.37	11.33	286.36	270.08
9	Maidenhair fern	Water	10	15.66	4.03	60.30	39.98
10	Helichrysum	EtOH 60%	55	48.63	18.57	210.55	134.05
11	Maidenhair fern	UBAE	100	43.96	8.87	311.87	219.43
12	Helichrysum	Water	10	17.68	10.58	96.44	67.95
13	Chelidonium	EtOH 60%	10	8.77	2.74	35.10	43.95
14	Pennyroyal	UBAE	55	22.91	10.00	352.37	240.99
15	Helichrysum	UBAE	100	47.03	27.93	181.46	179.56
16	Artemisia	EtOH 60%	55	36.41	9.77	228.76	103.84
17	Maidenhair fern	UBAE	10	30.58	5.72	227.36	75.04
18	Artemisia	UBAE	100	14.76	9.12	147.12	117.94
19	Artemisia	Water	10	19.97	6.12	130.73	100.54
20	Helichrysum	UBAE	10	18.19	11.52	68.31	39.04
21	Artemisia	Water	100	18.59	4.96	178.11	181.41
22	Pennyroyal	EtOH 60%	100	28.14	11.12	394.76	320.22
23	Maidenhair fern	Water	100	28.84	9.74	110.70	128.96

**Table 2 ijms-27-00309-t002:** ANOVA for reduced quadratic RSM models of TPC, TFC, FRAP, and DPPH.

Response	Source	DF	Sum of Squares	Mean Square	F-Ratio	*p*-Value	R^2^	Adj. R^2^	RMSE	PRESS	PRESS RMSE	PRESS RSquare (Q^2^)
TPC	Model	10	3447.20	344.72	8.354	0.0005 *	0.8744	0.7697	6.42	6070.6	16.25	0.54
	Error	12	495.16	41.26								
	Lack of fit	7	399.18	57.03	2.971	0.1244						
	Pure error	5	95.98	19.20								
TFC	Model	8	1046.83	130.85	15.052	<0.0001 *	0.9098	0.8609	1.49	266.88	3.41	0.64
	Error	14	121.77	8.70								
	Lack of fit	9	101.06	11.23	1.945	0.2701						
	Pure error	5	20.71	4.14								
FRAP	Model	9	313,257.20	34,806.36	24.896	<0.0001 *	0.9491	0.9165	39.19	52,079.2	47.58	0.74
	Error	13	18,185.04	1398.85								
	Lack of fit	8	16,485.50	2060.69	4.204	0.0953						
	Pure error	5	1699.54	339.91								
DPPH	Model	9	184,071.10	20,452.34	25.022	<0.0001 *	0.9495	0.9174	15.13	11,880.8	22.73	0.93
	Error	13	10,629.93	817.69								
	Lack of fit	8	9458.19	1182.27	4.512	0.0806						
	Pure error	5	1171.74	234.35								

* Significant model *p*-values (<0.05) are marked with an asterisk; DF: degrees of freedom; R^2^ and adjusted R^2^ are from the model summary for each response; lack-of-fit *p* > 0.05 indicates no significant lack of fit; RMSE: Root Mean Square Error; PRESS: Predicted Residual Sum of Squares.

**Table 3 ijms-27-00309-t003:** Variable importance in projection (VIP) scores from the PLS model relating extraction factors to TPC, TFC, FRAP, and DPPH. Predictors with VIP > 1 are considered highly influential.

Predictor Terms	Description *	VIP Score
*X*_1_{Pennyroyal}	Herb factor contrast: Pennyroyal vs. other herbs	1.98
*X*_1_{Helichrysum}	Herb factor contrast: Helichrysum vs. other herbs	1.86
*X* _3_	Solvent-to-solid ratio (coded)	1.72
*X* _3_ ^2^	Quadratic term for coded solvent-to-solid ratio	1.59
*X*_1_{Pennyroyal} × *X*_3_	Interaction: Pennyroyal × ratio	1.45
*X* _2_	Extraction technique (ordinal code: 1 = water; 2 = EtOH 60%; 3 = UBAE)	1.32
*X* _2_ ^2^	Quadratic term for extraction technique	1.28
*X*_1_{Chelidonium and Artemisia}	Herb factor contrast: Chelidonium and Artemisia vs. others	1.17
*X*_1_{Chelidonium and Artemisia} × *X*_2_	Interaction: Chelidonium and Artemisia × technique	1.08
*X*_1_{Pennyroyal} × *X*_2_	Interaction: Pennyroyal × technique	1.03

* Herb identity coded via contrasts.

**Table 4 ijms-27-00309-t004:** Comparison of the RSM composite optimum and the PLS compromise optimum for the extraction of phenolic-rich and antioxidant-rich herbal extracts. Predictors with VIP > 1 are considered highly influential.

Optimization Approach	RSM Composite Optimum *	PLS Compromise Optimum
*X*_1_—Herb	Helichrysum (TPC/TFC)/Pennyroyal (FRAP/DPPH) ^1^	Pennyroyal
*X*_2_—Technique	EtOH 60% (TPC/TFC)/ UBAE (FRAP/DPPH)	EtOH 60%
*X*_3_—Ratio (mL/g)	70–100	55
TPC (mg GAE/g dw)	60.32 ± 11.51	37.54 ± 2.10
TFC (mg RtE/g dw)	28.88 ± 2.92	21.62 ± 1.15
FRAP (µmol AAE/g dw)	465.26 ± 58.66	334.38 ± 12.37
DPPH (µmol AAE/g dw)	389.77 ± 29.01	262.67 ± 9.46
Composite desirability	>0.94 (per individual model)	0.6474

* Entries represent single-response optima from RSM; RSM composite optimum maximizes each response independently, producing different “best” settings for different metrics. ^1^ Infeasible as a single physical extraction without compromise—different herbs/techniques are required for each maximum.

**Table 5 ijms-27-00309-t005:** Polyphenolic composition of the PLS-optimum extract (Pennyroyal, EtOH 60%, solvent-to-solid ratio 55) was determined by HPLC-DAD. Bioactivities are sourced from published literature and not experimentally verified in this study.

Peak No.	Compound Name	Class	C (mg/L)	C (mg/g dw)	Reported Bioactivity	References
1	Protocatechuic acid	Phenolic acid	28.7 ± 1.2	1.78 ± 0.07	Antioxidant, anti-inflammatory	[35]
2	Neochlorogenic acid	Phenolic acid	6.59 ± 0.16	0.41 ± 0.01	Antioxidant, hepatoprotective	[36]
3	Chlorogenic acid	Phenolic acid	3.51 ± 0.18	0.22 ± 0.01	Antioxidant, hepatoprotective	[37]
4	Vanillic acid	Phenolic acid	0.42 ± 0.03	0.03 ± 0	Antioxidant	[38]
5	Caffeic acid	Phenolic acid	0.12 ± 0	0.01 ± 0	Antioxidant, antimicrobial	[39]
6	Syringic acid	Phenolic acid	1.94 ± 0.06	0.12 ± 0	Antioxidant	[40]
7	Kaempferol-3-*O*-β-rutinoside	Flavonoid glycoside	52.3 ± 1.1	3.23 ± 0.07	Antioxidant, cardioprotective	[41]
8	Quercetin-3-*D*-galactoside	Flavonoid glycoside	2.83 ± 0.06	0.17 ± 0	Antioxidant, anti-inflammatory	[42]
9	Luteolin-7-*O*-glucoside	Flavonoid glycoside	24.8 ± 0.7	1.53 ± 0.04	Antioxidant, anti-inflammatory	[43]
10	Kaempferol-3-glucoside	Flavonoid glycoside	6.91 ± 0.44	0.43 ± 0.03	Antioxidant	[44]
11	Isoharmetin-3-*O*-glucoside	Flavonoid glycoside	14.6 ± 0.4	0.90 ± 0.02	Antioxidant	[45]
12	Apigenin-7-*O*-glucoside	Flavonoid glycoside	0.94 ± 0.03	0.06 ± 0	Antioxidant, anti-inflammatory	[46]
13	Myricetin	Flavonoid aglycone	4.03 ± 0.17	0.25 ± 0.01	Antioxidant, anti-cancer	[47]
14	Rosmarinic acid	Phenolic acid	6.95 ± 0.47	0.43 ± 0.03	Strong antioxidant, anti-inflammatory	[48]
15	Quercetin	Flavonoid aglycone	2.32 ± 0.13	0.14 ± 0.01	Antioxidant, anti-cancer	[49]
16	Apigenin	Flavonoid aglycone	1.26 ± 0.07	0.08 ± 0	Antioxidant, anti-inflammatory	[50]
17	Kaempferol	Flavonoid aglycone	4.52 ± 0.24	0.28 ± 0.02	Antioxidant, anti-cancer	[51]
18	Rosmanol	Diterpene	2.62 ± 0.13	0.16 ± 0.01	Antioxidant, antimicrobial	[52]
	Total identified		165.4 ± 5.5	10.22 ± 0.34		

## Data Availability

The original contributions presented in this study are included in the article, and further inquiries can be directed to the corresponding author.

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
