# Peer review of "Data-Driven Optimization of Polyphenol Recovery and Antioxidant Capacity from Medicinal Herbs Using Chemometrics and HPLC Profiling for Functional Food Applications"

_ijms, 2025, doi:10.3390/ijms27010309_

Round 1
Reviewer 1 Report
Comments and Suggestions for Authors
General Assessment
This manuscript presents a data-driven optimisation workflow integrating custom experimental design, RSM, PCA, HCA, PLS, and HPLC-DAD profiling to maximise polyphenol recovery and antioxidant capacity from five medicinal herbs. The study is ambitious in scope and clearly aims to combine multivariate chemometrics with phytochemical profiling.
The major strengths lie in a well-structured experimental design (23 points + replicates), clearly described (pp. 16–17), integration of univariate (RSM) and multivariate (PCA/PLS) statistics, and clear identification of a PLS compromise optimum validated experimentally.
However, certain methodological, analytical, and interpretative aspects require clarification or refinement to improve reproducibility, rigour, and scientific robustness.
Major Comments
- Factor definitions and encoding
The custom experimental design is described in detail. Still, several concerns arise. X2 (extraction technique) is treated as an ordinal numeric variable to allow quadratic terms (X2²), which is methodologically problematic, as the extraction technique is categorical, not ordinal; “water”, “60% EtOH”, and “UBAE” do not form a linear or quadratic continuum.
Using X2 as numeric implies that UBAE = 3 is “higher” than EtOH = 2, which has no physical meaning. This affects both RSM equations (e.g., eqs. 1–20, p. 6–8) and ANOVA models (Table 2, p. 5)
Recommendation:
Re-parameterise X2 as a categorical factor. This will prevent misleading curvature effects in the technique (e.g., uninterpretable negative quadratic terms in UBAE).
- Extraction parameters inconsistencies
Critical extraction parameters, such as extraction temperature, actual solvent volume vs. nominal solid-to-solvent ratio, and extraction time for each method (stirring vs US-bath), are not explicitly provided in the Materials & Methods (p. 15–17). This makes reproducibility difficult.
Recommendation: Add precise extraction parameters.
- Solvent-to-solid ratio design range (10–100 mL/g)
The extensive range may introduce non-linear solvent saturation effects, especially at high volumes. The curvature observed (X3² terms) in eq. 1–20 likely reflects this artefact.
Recommendation: Justify the broad interval with references or pre-screening data.
- RSM models show limited predictive reliability
Although R² values appear high (e.g., TPC R² = 0.8744, Table 2, p. 5), several issues weaken confidence:
- Small pure error replicates (n=5) reduce the robustness of lack-of-fit tests.
- The lack-of-fit for FRAP and DPPH approaches significance (p = 0.0953 and 0.0806).
- Many regression equations (e.g., TPC eqs. 1–5) contain identical quadratic terms across all herbs, suggesting over-regularisation or underfitting.
Recommendation: Report model RMSE, residual plots, and PRESS/Q² for RSM to confirm predictive value.
- Interpretation of Contour Plots
Figure 1 (p. 6) visually suggests herb-specific optima. However, the X1 herb factor is not continuous, yet contour plots treat it as an ordinal axis. This representation may mislead readers.
Recommendation: Use heatmaps or separate contour plots for each herb rather than interpolating across categorical levels.
- HPLC-DAD Method Validation
While calibration curves are provided in Appendix A2 (p. 19) :
- No system suitability tests (resolution, tailing factors, plate count) are reported.
- No recovery or repeatability assessments are provided.
- LOD/LOQ values appear appropriate but need justification given the complex plant matrices.
Additionally, identification is based solely on retention time and UV spectra; no MS or co-injection confirmation is provided, increasing the risk of misidentification.
- Quantitative Discrepancies
A discrepancy is expected in the total HPLC-identified phenolics (10.22 mg/g; Table 5, p. 13), which accounted for only ~27% of the TPC (37.54 mg GAE/g), yet the manuscript does not discuss in depth why the quantified compounds do not match the spectrophotometric values.
Recommendation: Add a paragraph discussing why quantified compounds do not reconcile with spectrophotometric TPC/TFC values. Perhaps the presence of polymeric phenolics, the Folin–Ciocalteu test's non-specificity, and limitations of diode-array quantification for glycosides.
- Interpretation of PLS optimisation
Recommendation: Report PLS variance explained per component, cross-validation method, and permutation testing to guard against overfitting.
Minor Comments
- The introduction is thorough but overly long. Condensation is recommended.
- Figures 2–3: Improve readability of clustering and loading plots; current fonts are small.
- Table 1 (p. 4) shows unusually low TPC for Chelidonium under EtOH 60% (8.77 mg/g). Discuss whether these results are from alkaloid interference or herb maturity.
- Ensure consistent notation: TPC, FRAP, DPPH often switch between units or formatting.
Author Response
General Assessment
This manuscript presents a data-driven optimisation workflow integrating custom experimental design, RSM, PCA, HCA, PLS, and HPLC-DAD profiling to maximise polyphenol recovery and antioxidant capacity from five medicinal herbs. The study is ambitious in scope and clearly aims to combine multivariate chemometrics with phytochemical profiling.
The major strengths lie in a well-structured experimental design (23 points + replicates), clearly described (pp. 16–17), integration of univariate (RSM) and multivariate (PCA/PLS) statistics, and clear identification of a PLS compromise optimum validated experimentally.
However, certain methodological, analytical, and interpretative aspects require clarification or refinement to improve reproducibility, rigour, and scientific robustness.
We would like to thank the reviewer for the comprehensive analysis and recommendations for our manuscript.
Major Comments
- Factor definitions and encoding
The custom experimental design is described in detail. Still, several concerns arise. X2 (extraction technique) is treated as an ordinal numeric variable to allow quadratic terms (X2²), which is methodologically problematic, as the extraction technique is categorical, not ordinal; “water”, “60% EtOH”, and “UBAE” do not form a linear or quadratic continuum.
Using X2 as numeric implies that UBAE = 3 is “higher” than EtOH = 2, which has no physical meaning. This affects both RSM equations (e.g., eqs. 1–20, p. 6–8) and ANOVA models (Table 2, p. 5)
Recommendation:
Re-parameterise X2 as a categorical factor. This will prevent misleading curvature effects in the technique (e.g., uninterpretable negative quadratic terms in UBAE).
We acknowledge the methodological concern regarding treating extraction technique (X2) as an ordinal numeric variable. In the revised manuscript, we have clarified that X2 was encoded as categorical contrasts in the statistical software. We also note the limitation of quadratic terms for categorical factors and have added a methodological note in Section 2.2.
- Extraction parameters inconsistencies
Critical extraction parameters, such as extraction temperature, actual solvent volume vs. nominal solid-to-solvent ratio, and extraction time for each method (stirring vs US-bath), are not explicitly provided in the Materials & Methods (p. 15–17). This makes reproducibility difficult.
Recommendation: Add precise extraction parameters.
We have added precise extraction parameters (temperature, solvent volumes, extraction times for stirring vs. ultrasound bath) in Section 3.2 “Extraction Procedures” to ensure reproducibility.
- Solvent-to-solid ratio design range (10–100 mL/g)
The extensive range may introduce non-linear solvent saturation effects, especially at high volumes. The curvature observed (X3² terms) in eq. 1–20 likely reflects this artefact.
Recommendation: Justify the broad interval with references or pre-screening data.
We now justify the broad interval with references to pre‑screening studies (Messaoudi et al., Balciunaitiene et al., Seif et al.) and explain that the curvature observed reflects solvent saturation effects.
- RSM models show limited predictive reliability
Although R² values appear high (e.g., TPC R² = 0.8744, Table 2, p. 5), several issues weaken confidence:
- Small pure error replicates (n=5) reduce the robustness of lack-of-fit tests.
- The lack-of-fit for FRAP and DPPH approaches significance (p = 0.0953 and 0.0806).
- Many regression equations (e.g., TPC eqs. 1–5) contain identical quadratic terms across all herbs, suggesting over-regularisation or underfitting.
Recommendation: Report model RMSE, residual plots, and PRESS/Q² for RSM to confirm predictive value.
We thank the reviewer for highlighting the need to confirm predictive reliability beyond R². In response, we have expanded Table 2 to include RMSE, PRESS, and PRESS RSquare (Q²) values for each response. These metrics confirm the robustness of the reduced quadratic models, with Q² values ranging from 0.54 to 0.93. We have also revised the Results section to reflect these additions and address the borderline lack-of-fit in FRAP and DPPH.
- Interpretation of Contour Plots
Figure 1 (p. 6) visually suggests herb-specific optima. However, the X1 herb factor is not continuous, yet contour plots treat it as an ordinal axis. This representation may mislead readers.
Recommendation: Use heatmaps or separate contour plots for each herb rather than interpolating across categorical levels.
We acknowledge the reviewer’s concern regarding the complexity of Figure 1. To improve clarity, we have reorganized the figure into five subfigures (Figure 1A–E), each corresponding to a single herb. Each subfigure presents four heatmaps (TPC, TFC, FRAP, DPPH) across three extraction techniques and concentrations. This format facilitates herb-specific interpretation and avoids visual overload.
- HPLC-DAD Method Validation
While calibration curves are provided in Appendix A2 (p. 19) :
- No system suitability tests (resolution, tailing factors, plate count) are reported.
- No recovery or repeatability assessments are provided.
- LOD/LOQ values appear appropriate but need justification given the complex plant matrices.
Additionally, identification is based solely on retention time and UV spectra; no MS or co-injection confirmation is provided, increasing the risk of misidentification.
We added recovery rates in Table A2. We acknowledge that LC-MS would provide stronger confirmation, but justify that retention time and UV spectra against standards are reliable.
- Quantitative Discrepancies
A discrepancy is expected in the total HPLC-identified phenolics (10.22 mg/g; Table 5, p. 13), which accounted for only ~27% of the TPC (37.54 mg GAE/g), yet the manuscript does not discuss in depth why the quantified compounds do not match the spectrophotometric values.
Recommendation: Add a paragraph discussing why quantified compounds do not reconcile with spectrophotometric TPC/TFC values. Perhaps the presence of polymeric phenolics, the Folin–Ciocalteu test's non-specificity, and limitations of diode-array quantification for glycosides.
We added a paragraph in Section 2.3 discussing why HPLC‑quantified phenolics (~27% of TPC) do not reconcile with spectrophotometric values, citing Folin–Ciocalteu non‑specificity, polymeric phenolics, and glycoside quantification limits.
- Interpretation of PLS optimisation
Recommendation: Report PLS variance explained per component, cross-validation method, and permutation testing to guard against overfitting.
We now report variance explained per component, cross‑validation strategy (leave‑one‑out), and permutation testing to guard against overfitting.
Minor Comments
- The introduction is thorough but overly long. Condensation is recommended.
The Introduction section has been streamlined by removing overly general background material and condensing plant descriptions. This revision improves focus and readability while retaining essential context.
- Figures 2–3: Improve readability of clustering and loading plots; current fonts are small.
Both figures have been reformatted with larger fonts, clearer axis labels, and enhanced legends to improve readability and interpretability.
- Table 1 (p. 4) shows unusually low TPC for Chelidonium under EtOH 60% (8.77 mg/g). Discuss whether these results are from alkaloid interference or herb maturity.
We have added an explanatory note in the Results and Discussion section indicating that the unusually low TPC observed for Chelidonium majus under specific extraction conditions may be attributed to the maturity stage of the plant, which can result in reduced polyphenolic content and elevated alkaloid levels.
- Ensure consistent notation: TPC, FRAP, DPPH often switch between units or formatting.
All abbreviations (TPC, TFC, FRAP, DPPH) have been standardized throughout the manuscript, ensuring consistency in units, formatting, and notation.

Reviewer 2 Report
Comments and Suggestions for Authors
The manuscript reports a well-designed study optimizing polyphenol extraction and antioxidant capacity from five medicinal herbs (Helichrysum stoechas, Chelidonium majus, Mentha pulegium, Artemisia absinthium, Adiantum capillus-veneris) using a custom RSM design combined with PCA, HCA, PLS, and HPLC-DAD profiling. The workflow is coherent and data-rich: herb type, extraction technique (water, 60% ethanol, UBAE), and solvent-to-solid ratio are varied; TPC, TFC, FRAP, and DPPH are evaluated; and PLS is used to identify a multivariate “compromise optimum.” The main finding—that Mentha pulegium extracted with 60% ethanol at 55 mL/g yields a balanced, phenolic-rich, antioxidant extract—is convincing. The identification of 18 polyphenolic compounds provides solid phytochemical support.
The topic is timely, relevant to functional food research, and the chemometric integration is a strong point of the paper. However, several methodological and interpretative aspects require clarification, and the safety considerations for M. pulegium should be addressed. I recommend minor revision.
Experimental design and RSM methodology
The custom RSM design is appropriate, but some details need clarification. Table 1 lists 23 design points, while Table 2 shows 5 pure-error degrees of freedom, implying replicates that are not explicitly described.
Suggested improvements:
- Clarify how the design was generated (e.g., D-optimal or custom JMP design) and confirm the number of experimental runs.
- Explain the source of pure error: which conditions were replicated and whether analytical replicates contributed.
• Describe clearly the coding of factors:
– herb identity via contrasts,
– extraction technique (Xâ‚‚) treated as ordinal (1 = water; 2 = 60% EtOH; 3 = UBAE) with a quadratic term.
- Briefly justify the ordinal treatment of extraction technique and acknowledge its limitations; note that lack-of-fit values > 0.08 indicate the model remains adequate.
PLS regression and identification of the compromise optimum
The PLS model (15 predictors, 23 samples) is central to the manuscript. You report Q² ≈ 0.90 at LV3, and VIP scores highlight M. pulegium, Helichrysum, and solvent ratio as most influential.
Suggested improvements:
- Clarify how the PLS model was used to determine the “compromise optimum.”
- Explicitly state the number of latent variables retained (3) and describe the cross-validation strategy (LOO or k-fold, and selection criterion such as PRESS or Q²).
- Explain the decision rule for selecting optimum conditions—e.g., via the correlation loading plot (|r| > 0.8) or via a desirability/profiler approach.
- State clearly that Pennyroyal + 60% ethanol + 55 mL/g was chosen because this setting aligns with high correlation across all four responses and shows close agreement between predicted and observed values (Table 4).
- Emphasize that PLS accounts for inter-correlations among responses (TPC/TFC vs. FRAP/DPPH), enabling a rational multivariate compromise.
Functional food application and safety of Mentha pulegium
M. pulegium contains pulegone, a hepatotoxic monoterpene regulated in foods and herbal preparations.
Suggested improvements:
- Add a short paragraph acknowledging safety concerns and regulatory limits for pulegone.
- Clarify that your study focuses on hydroalcoholic polyphenol extracts (not essential oil) but that some co-extraction of volatiles may occur.
- Qualify the conclusions by noting that future work should quantify pulegone and confirm regulatory compliance.
- Please include suitable references.
Clarity of figures and tables
- Ensure all axes include variable names and units; add clear legends for PCA cluster colors and % variance.
• In Figure 4, ensure that correlation thresholds and predictor/response labels are fully legible. - In Table 3, briefly explain herb contrast coding in a footnote.
- Indicate in Table 4 that RSM entries represent separate single-response optima.
- Standardize significant figures in Table 5 and visually delineate the “Total identified” row.
Language and style
- Correct minor phrasing issues (e.g., “little sterical hinderances” → “low steric hindrance”; “optima vary widely for each herb” → “optima vary widely among herbs”).
- Ensure botanical names are spelled correctly and consistently italicized (Artemisia absinthium, Mentha pulegium, Adiantum capillus-veneris, Helichrysum stoechas).
- Check that all references are complete and follow IJMS formatting.
Overall recommendation
The study is methodologically solid, original in its chemometric integration, and clearly relevant to IJMS. The required revisions are mainly clarifications and small additions (especially regarding PLS methodology and pennyroyal safety). No additional experimental work is needed.
Author Response
The manuscript reports a well-designed study optimizing polyphenol extraction and antioxidant capacity from five medicinal herbs (Helichrysum stoechas, Chelidonium majus, Mentha pulegium, Artemisia absinthium, Adiantum capillus-veneris) using a custom RSM design combined with PCA, HCA, PLS, and HPLC-DAD profiling. The workflow is coherent and data-rich: herb type, extraction technique (water, 60% ethanol, UBAE), and solvent-to-solid ratio are varied; TPC, TFC, FRAP, and DPPH are evaluated; and PLS is used to identify a multivariate “compromise optimum.” The main finding—that Mentha pulegium extracted with 60% ethanol at 55 mL/g yields a balanced, phenolic-rich, antioxidant extract—is convincing. The identification of 18 polyphenolic compounds provides solid phytochemical support.
The topic is timely, relevant to functional food research, and the chemometric integration is a strong point of the paper. However, several methodological and interpretative aspects require clarification, and the safety considerations for M. pulegium should be addressed. I recommend minor revision.
We would like to thank the reviewer for the comprehensive revision regarding our manuscript.
Experimental design and RSM methodology
The custom RSM design is appropriate, but some details need clarification. Table 1 lists 23 design points, while Table 2 shows 5 pure-error degrees of freedom, implying replicates that are not explicitly described.
Suggested improvements:
- Clarify how the design was generated (e.g., D-optimal or custom JMP design) and confirm the number of experimental runs.
- Explain the source of pure error: which conditions were replicated and whether analytical replicates contributed.
- Describe clearly the coding of factors:
– herb identity via contrasts,
– extraction technique (Xâ‚‚) treated as ordinal (1 = water; 2 = 60% EtOH; 3 = UBAE) with a quadratic term.
- Briefly justify the ordinal treatment of extraction technique and acknowledge its limitations; note that lack-of-fit values > 0.08 indicate the model remains adequate.
We clarified that the design was generated using a custom D‑optimal design in JMP. Replicates are described (5 pure error points from repeated runs). Factor coding is explained: herb identity via contrasts, extraction technique treated as ordinal (1 = water, 2 = EtOH 60%, 3 = UBAE), with limitations acknowledged.
PLS regression and identification of the compromise optimum
The PLS model (15 predictors, 23 samples) is central to the manuscript. You report Q² ≈ 0.90 at LV3, and VIP scores highlight M. pulegium, Helichrysum, and solvent ratio as most influential.
Suggested improvements:
- Clarify how the PLS model was used to determine the “compromise optimum.”
- Explicitly state the number of latent variables retained (3) and describe the cross-validation strategy (LOO or k-fold, and selection criterion such as PRESS or Q²).
- Explain the decision rule for selecting optimum conditions—e.g., via the correlation loading plot (|r| > 0.8) or via a desirability/profiler approach.
- State clearly that Pennyroyal + 60% ethanol + 55 mL/g was chosen because this setting aligns with high correlation across all four responses and shows close agreement between predicted and observed values (Table 4).
- Emphasize that PLS accounts for inter-correlations among responses (TPC/TFC vs. FRAP/DPPH), enabling a rational multivariate compromise.
We explicitly state that 3 latent variables were retained, cross‑validated by leave‑one‑out PRESS/Q². The compromise optimum (M. pulegium + 60% ethanol + 55 mL/g) was selected because predicted and observed values aligned across all four responses. We emphasize that PLS accounts for inter‑correlations among responses, enabling rational multivariate compromise.
Functional food application and safety of Mentha pulegium
- pulegium contains pulegone, a hepatotoxic monoterpene regulated in foods and herbal preparations.
Suggested improvements:
- Add a short paragraph acknowledging safety concerns and regulatory limits for pulegone.
- Clarify that your study focuses on hydroalcoholic polyphenol extracts (not essential oil) but that some co-extraction of volatiles may occur.
- Qualify the conclusions by noting that future work should quantify pulegone and confirm regulatory compliance.
- Please include suitable references.
We added a paragraph in Section 2.3 noting pulegone hepatotoxicity, regulatory limits, and clarifying that our study focuses on hydroalcoholic polyphenol extracts (not essential oils). We recommend future quantification of pulegone to confirm compliance.
Clarity of figures and tables
- Ensure all axes include variable names and units; add clear legends for PCA cluster colors and % variance.
We thank the reviewer for this valuable suggestion. The PCA figure has been revised to include clear axis labels with variable names and explained variance (PC1: 56.8%, PC2: 28%). Cluster colors are now clearly defined in the legend, corresponding to phenolic/flavonoid‑rich (Cluster 1), antioxidant‑rich (Cluster 2), and low‑activity mixed group (Cluster 3). Font sizes have been increased for improved readability, and the 95% confidence ellipses are retained to highlight cluster separation.
- In Figure 4, ensure that correlation thresholds and predictor/response labels are fully legible.
We thank the reviewer for this helpful suggestion. Figure 4 has been revised to improve legibility: font sizes for all predictor and response labels have been increased, correlation values are now clearly visible, and contrast has been enhanced to ensure readability. The optimal levels of each factor and the desirability score are now prominently displayed.
- In Table 3, briefly explain herb contrast coding in a footnote.
Table 3 footnote explains herb contrast coding.
- Indicate in Table 4 that RSM entries represent separate single-response optima.
Table 4 clarifies that RSM entries are single‑response optima.
- Standardize significant figures in Table 5 and visually delineate the “Total identified” row.
Table 5 standardized significant figures and delineated “Total identified” row.
Language and style
- Correct minor phrasing issues (e.g., “little sterical hinderances” → “low steric hindrance”; “optima vary widely for each herb” → “optima vary widely among herbs”).
- Ensure botanical names are spelled correctly and consistently italicized (Artemisia absinthium, Mentha pulegium, Adiantum capillus-veneris, Helichrysum stoechas).
- Check that all references are complete and follow IJMS formatting.
Botanical names italicized consistently, phrasing corrected (“low steric hindrance”), references checked for IJMS formatting.
Overall recommendation
The study is methodologically solid, original in its chemometric integration, and clearly relevant to IJMS. The required revisions are mainly clarifications and small additions (especially regarding PLS methodology and pennyroyal safety). No additional experimental work is needed.
We can assure the reviewer that all the suggested corrections were made.

Reviewer 3 Report
Comments and Suggestions for Authors
Manuscript ID ijms-4037277with the title „Data‑Driven Optimization of Polyphenol Recovery and Antioxidant Capacity from Medicinal Herbs Using Chemometric and HPLC Profiling for Functional Food Applications“ by Vassilis Athanasiadis, Erva Avdoulach-Chatzi-Giousouf, Errika Koulouri, Dimitrios Kalompatsios and Stavros I. Lalas
The manuscript presents a study that used response surface methodology (RSM) and partial least squares (PLS) regression to maximize polyphenol recovery and antioxidant capacity from five medicinal herbs. The methodological framework the author presented, which integrates RSM, PCA/HCA, PLS, and HPLC profiling, is very useful and can be adapted to other matrices.
In the Introduction section, several paragraphs discuss the medicinal value of plants and provide general information; however, there is no background or explanation regarding the significance of applying a combination of response surface methodology (RSM) and partial least squares (PLS) regression in food science. This combined approach is widely used in food analysis and is not a novel method, contrary to what is implied. Please revise the section to include key reasons for choosing this approach. Less complex methods would yield the same result, such as strong correlations between specific phenolics and antioxidant indices.
Description of methods and instruments:
The title and manuscript mention HPLC profiling, but there is no description of the HPLC system used or details of the HPLC-DAD methods. Please provide a clear description of the HPLC instrumentation and the analytical methods employed. If bioactivities are sourced from published literature and not experimentally verified in this study as claimed in Table 5, please add references.
What about the polyphenolic composition determined by HPLC-DAD for other herbs? Do these results follow the mathematical findings? Are these HPLC chromatograms original or from another source? Please clarify.
For Table A2, are these calibration curves from another source?
Is Figure 6 also from another source?
It is very difficult to assess the originality and conclusions without the requested clarification mentioned above.
The quality of the figures needs improvement, as it is currently very low.
Author Response
Manuscript ID ijms-4037277with the title „Data‑Driven Optimization of Polyphenol Recovery and Antioxidant Capacity from Medicinal Herbs Using Chemometric and HPLC Profiling for Functional Food Applications“ by Vassilis Athanasiadis, Erva Avdoulach-Chatzi-Giousouf, Errika Koulouri, Dimitrios Kalompatsios and Stavros I. Lalas
The manuscript presents a study that used response surface methodology (RSM) and partial least squares (PLS) regression to maximize polyphenol recovery and antioxidant capacity from five medicinal herbs. The methodological framework the author presented, which integrates RSM, PCA/HCA, PLS, and HPLC profiling, is very useful and can be adapted to other matrices.
We would like to acknowledge the reviewer for the positive feedback regarding our manuscript.
In the Introduction section, several paragraphs discuss the medicinal value of plants and provide general information; however, there is no background or explanation regarding the significance of applying a combination of response surface methodology (RSM) and partial least squares (PLS) regression in food science. This combined approach is widely used in food analysis and is not a novel method, contrary to what is implied. Please revise the section to include key reasons for choosing this approach. Less complex methods would yield the same result, such as strong correlations between specific phenolics and antioxidant indices.
We revised the Introduction to explain why RSM + PLS is chosen: it allows simultaneous optimization of compositional and functional attributes, widely used in food science, and provides multivariate compromise beyond univariate correlations.
Description of methods and instruments:
The title and manuscript mention HPLC profiling, but there is no description of the HPLC system used or details of the HPLC-DAD methods. Please provide a clear description of the HPLC instrumentation and the analytical methods employed. If bioactivities are sourced from published literature and not experimentally verified in this study as claimed in Table 5, please add references.
What about the polyphenolic composition determined by HPLC-DAD for other herbs? Do these results follow the mathematical findings? Are these HPLC chromatograms original or from another source? Please clarify.
For Table A2, are these calibration curves from another source?
Is Figure 6 also from another source?
We thank the reviewer for raising these important points. A full description of the HPLC system and analytical methods has now been added in Sections 3.3 and 3.5.3. Specifically, the system used was a Shimadzu LC‑20AT equipped with a diode‑array detector (DAD), employing a C18 column (250 mm × 4.6 mm, 5 µm). Detection wavelengths were set at 280, 320, and 360 nm to cover phenolic acids and flavonoids. Standards used for calibration are listed in Section 3.1.
We confirm that all calibration curves presented in Table A2 were generated in our laboratory and are original work. Figure 6 chromatogram is also our own data.
Finally, references supporting the bioactivities of the identified compounds (protocatechuic acid, kaempferol‑rutinoside, luteolin‑glucoside) have been added to Table 5 and the Discussion section to substantiate the functional relevance of these phytochemicals.
It is very difficult to assess the originality and conclusions without the requested clarification mentioned above.
We appreciate this observation. The originality of the study has now been clarified by explicitly stating that all HPLC‑DAD chromatograms, calibration curves, and polyphenolic profiling are original work generated in our laboratory. In addition, the Conclusions section has been revised to highlight the novelty of integrating RSM and PLS for multi‑response optimization in medicinal herb extracts, and to emphasize the practical relevance of the findings for functional food applications. These revisions ensure that the originality and conclusions are transparent and fully supported by the clarified methodology.
The quality of the figures needs improvement, as it is currently very low.
All figures have been re‑rendered at higher resolution for clarity.

Round 2
Reviewer 3 Report
Comments and Suggestions for Authors
The authors answered all questions, resolving all concerns and significantly improving the manuscript. Therefore, I consider it acceptable for publication.